# Inhibitors of the Sialidase NEU3 as Potential Therapeutics for Fibrosis

**DOI:** 10.3390/ijms24010239

**Published:** 2022-12-23

**Authors:** Tejas R. Karhadkar, Wensheng Chen, Darrell Pilling, Richard H. Gomer

**Affiliations:** Department of Biology, Texas A&M University, College Station, TX 77843-3474, USA

**Keywords:** neuraminidase, idiopathic pulmonary fibrosis, desialylation, sialylation, transforming growth factor, serum amyloid P, pentraxin, biomarker

## Abstract

Fibrosing diseases are a major medical problem, and are associated with more deaths per year than cancer in the US. Sialidases are enzymes that remove the sugar sialic acid from glycoconjugates. In this review, we describe efforts to inhibit fibrosis by inhibiting sialidases, and describe the following rationale for considering sialidases to be a potential target to inhibit fibrosis. First, sialidases are upregulated in fibrotic lesions in humans and in a mouse model of pulmonary fibrosis. Second, the extracellular sialidase NEU3 appears to be both necessary and sufficient for pulmonary fibrosis in mice. Third, there exist at least three mechanistic ways in which NEU3 potentiates fibrosis, with two of them being positive feedback loops where a profibrotic cytokine upregulates NEU3, and the upregulated NEU3 then upregulates the profibrotic cytokine. Fourth, a variety of NEU3 inhibitors block pulmonary fibrosis in a mouse model. Finally, the high sialidase levels in a fibrotic lesion cause an easily observed desialylation of serum proteins, and in a mouse model, sialidase inhibitors that stop fibrosis reverse the serum protein desialylation. This then indicates that serum protein sialylation is a potential surrogate biomarker for the effect of sialidase inhibitors, which would facilitate clinical trials to test the exciting possibility that sialidase inhibitors could be used as therapeutics for fibrosis.

## 1. Fibrosis

Fibrosis is a disease that involves an increased amount of scar tissue appearing in an internal organ. In a fibrosing disease, an insult or an injury to an internal organ initiates an uncontrolled wound repair mechanism which leads to a progressive increase in the deposition of scar tissue, eventually leading to organ failure and death [1,2]. There are more than 60 different fibrosing diseases, and 30–45% of deaths in the western world are from diseases where fibrosis was a significant factor [1]. One tissue that can develop fibrosis is the lung [3]. The lungs are directly exposed to external harmful environmental factors present in air, which can cause a recurring injury in the lungs, initiating a repeated wound healing response and eventually causing pulmonary fibrosis [4,5,6,7].

Interstitial lung disease (ILD) is a general term for a group of diseases that includes idiopathic pulmonary fibrosis (IPF), a chronic inflammatory disorder characterized by the accumulation of scar tissue in the lungs [8]. With an incidence rate of 2–30 cases per 100,000 person-years, this translates to a population prevalence of ~3 million people worldwide [8,9]. IPF has a survival rate of only 30% within five years after initial diagnosis, and has an incidence of 1 in 400 in patients older than 65 years [10]. Despite the high prevalence of IPF, there are only two FDA-approved therapeutics which slow down but do not reverse the progression of the disease [11].

## 2. Sialidases

In eukaryotic cells, some proteins undergo post-translational modification, such as glycosylation adding a chain of sugar molecules onto specific amino acids in the protein sequence [12,13]. In some glycoconjugates, the distal tip or tips (some glycoconjugates are branched) is the sugar sialic acid [14,15,16]. The sialic acid can make a glycoconjugate fully functional and can thus affect physiological and pathological processes [16,17,18,19]. Conversely, loss of the sialic acid can cause the glycoconjugate to lose its functional ability [16,17,20].

The enzymes that remove sialic acid from glycoconjugates are called neuraminidases or sialidases. Sialidases are present in, and sometimes on, viruses, bacteria, protozoa, and mammalian cells [21,22]. Viruses such as influenza require sialidase to release the virus from the sialic acids on the outside of a host cell [23]. The bacterial respiratory pathogen *Pseudomonas aeruginosa* uses a sialidase to colonize the lungs [24]. The sialidase NanA from *Streptococcus pneumoniae* causes downregulation of tight junction protein expression in endothelial cells and allows bacterial invasion through the blood–brain barrier, which can then affect the central nervous system [25]. Mammals have four sialidases, named NEU1, NEU2, NEU3, and NEU4. NEU1 is in the lysosome [26,27,28], NEU2 is a soluble, cytosolic enzyme, and NEU4 has 2 isoforms, one in mitochondria, and the other on intracellular membranes [29,30,31]. NEU3 is in endosomes and on the extracellular side of the plasma membrane, and under some conditions, can be released from the membrane to the extracellular environment [32].

Mammalian sialidases play an important role in the catabolism of glycoconjugates, modulation of the sialylation on cell membranes, skeletal muscle architecture, central nervous system function, and immune function [33,34,35]. Immune cells express different sialidases. Human neutrophils contain sialidase enzyme activity and express NEU2 protein [36]. As monocytes differentiate into macrophages, they increase the expression of NEU1 and NEU3 and reduce the expression of NEU4 [37], and T cells have increased NEU1 expression after stimulation [38].

Upregulation of sialidases is associated with inflammation [39,40,41,42,43,44,45]. For instance, NEU1 potentiates airway inflammation in asthma [46], and upregulated levels of NEU1 potentiate inflammation in atherosclerosis [47]. In cardiovascular disease models, both NEU1 and NEU3 are upregulated, and genetic disruption of *NEU1* or *NEU3* in mice, or treatment of mice with sialidase inhibitors, attenuate atherosclerosis, cardiac hypertrophy, or cardiac fibrosis [48,49,50]. In the LPS-induced air pouch model of inflammation, leukocyte recruitment is reduced in *Neu1^−/−^* and *Neu3^−/−^* mice, while it is increased in *Neu4^−/−^* mice [51]. *Neu3^−/−^* mice are also protected from *Salmonella enterica* Typhimurium-induced intestinal inflammation and colitis [52], and colitis-induced colon cancer [53]. NEU3 is upregulated in, and is important for the survival of, tumor cells in colon [54], renal [55], ovarian [56], and prostate cancers [57].

Sialidase inhibitors have been used to treat inflammation in both preclinical models and in patients. Zanamivir (Relenza) was designed to inhibit influenza sialidases, but at 100 mg/kg, it has shown efficacy in an animal model of arthritis, with reduced joint inflammation and arthritis score, reduced autoantibody production, and increased sialyation of serum IgG [58]. Recently, sialidase inhibitors (oseltamivir and peramivir) have been shown to reduce mortality in patients with severe COVID-19 [59].

In addition, the sialidase NEU1 is upregulated in IPF [60]. Although the specific upregulated sialidase was not identified, 8 of 9 IPF patients had a high sialidase activity in the lung bronchoalveolar lavage (BAL) fluid, while healthy controls showed no detectable sialidase activity [61]. These studies point towards a significant involvement of sialidases in the progression of pulmonary fibrosis.

## 3. Transforming Growth Factor-β1 (TGF-β1)

TGF-β1 is a homodimer peptide signal that regulates processes such as cell proliferation and growth, apoptosis, and maintaining immune homeostasis [62,63,64]. TGF-β1 upregulation plays a key role in the progression of pulmonary fibrosis [65]. TGF-β1 is synthesized as a precursor polypeptide consisting of latency-associated peptide (LAP) and what will become one of the peptides of the active TGF-β1 product [66]. The precursor polypeptide is processed in the endoplasmic reticulum (ER) within a cell and dimerizes to form a complex of two active TGF-β1 polypeptides enclosed in two LAPs [63]. For further processing, the nascent TGF-β1 is transferred from the ER to the Golgi, where it undergoes glycosylation/sialylation events on three asparagine residues, which are all present on LAP [63,66,67]. This processed complex is secreted from cells alone or in association with the latent TGF-β-binding protein (LTBP) [63,66]. Several mechanisms such as proteolysis, mechanical tension, or changes in pH cause LAP to release active TGF-β1 [66,68]. In addition to the above mechanisms, viral and bacterial sialidases can desialylate LAP causing LAP to change its conformation and releases active TGF-β1 [25,69,70,71,72,73].

The released active TGF-β1 plays an important role in maintaining homeostasis within the lungs [68,74,75]. When homeostasis is disturbed and TGF-β1 is upregulated, it induces epithelial-to-mesenchymal transition (EMT) in the alveolar epithelial cells [76,77], potentiates proliferation and differentiation of fibroblasts to myofibroblasts, and causes deposition of collagen in the surrounding tissue of the lungs [65,78,79,80]. Since these are a few of many hallmarks of pulmonary fibrosis, targeting TGF-β1 can be a potential therapeutic strategy to control pulmonary fibrosis. This strategy has been previously tried, but the strategies that directly block TGF-β1 cause various side effects, often interfering with the vital homeostatic functions of active TGF-β1 [81]. Thus, a better strategy needs to be devised to target the upregulation of TGF-β1 in a fibrotic lesion, rather than global TGF-β1.

## 4. NEU3 Is Necessary and Sufficient for Pulmonary Fibrosis and Is Elevated in Fibrosis in Other Organs

There is extensive desialylation of glycoconjugates and upregulation of the sialidase NEU3 in the fibrotic lesions of patients with pulmonary fibrosis and mice with bleomycin-induced pulmonary fibrosis [61,82,83,84,85]. For instance, we observed extensive desialylation of glycoconjugates and upregulation of the sialidases NEU1, 2, and 3 in fibrotic lesions in human and male mouse lungs. Most glycoconjugates are outside cells, and of the sialidases, only NEU3 is located on the extracellular side of the plasma membrane. Compared to male and female C57BL/6 mice, male and female *Neu3^−/−^* mice showed strongly attenuated bleomycin-induced weight loss, lung damage, inflammation, upregulation of TGF-β1, and fibrosis [83]. In the *Neu3^−/−^* mice, bleomycin did not upregulate NEU1 or NEU2, indicating that NEU3 is a key driver of sialidase upregulation [83]. Conversely, aspirations of NEU3 (with levels corresponding to the level of NEU3 in a fibrotic mouse lung) induced fibrosis in male and female mice [86]. These results suggest that NEU3 is necessary and sufficient for pulmonary fibrosis in mice. We found that epithelial cells, endothelial cells, macrophages, and fibroblasts in the fibrotic lungs express NEU3 [82,87], that NEU3, but not other sialidases, is upregulated in BAL fluid from pulmonary fibrosis patients and male and female mice at day 21 after bleomycin ([82,83], and observed elevated NEU3 in human fibrotic heart, kidney, and liver. Together, these results indicate that NEU3 potentiates fibrosis.

## 5. NEU3 Desialylates and Inactivates the Anti-Fibrotic Serum Protein SAP and SAP Is Desialylated in the Sera of IPF Patients

We previously found that a human serum glycoprotein called serum amyloid P (SAP) [88] calms the innate immune system [20,89,90,91,92], and that injections of SAP are therapeutic in mouse and rat models of pulmonary fibrosis [93]. We and others found that SAP injections are therapeutic in a wide variety of other fibrosis models [94,95,96,97,98,99,100,101,102,103,104,105]. Intravenous administration of recombinant SAP (called PRM-151 and zinpentraxin-alfa) was safe in Phase 1 trials [106,107] and works significantly better than the current standard of care in Phase 2 clinical trials for pulmonary fibrosis [108,109,110]. Phase 3 trials for SAP/PRM-151 in pulmonary fibrosis patients are ongoing. SAP is a homopentamer, and each monomer has a branched glycosylation with sialic acid at the distal tips of the glycosylation [88,111]. We found that SAP calms the innate immune system in large part by using its glycosylation to activate the lectin receptor DC-SIGN on innate immune system cells [20], and that desialylation of SAP blocks its ability to calm the innate immune system [20,111]. We found extensive desialylation of SAP in the sera of patients with idiopathic pulmonary fibrosis (IPF), compared to the SAP in the sera of healthy volunteers [111]. In the IPF patients, there was more extensive desialylation of SAP in progressive IPF cases than in stable IPF cases [111]. Compared to SAP from healthy donors, the SAP from IPF patients had a poor biological activity. Enzymatically sialylating the SAP from IPF patient sera made the SAP as active as control SAP, and enzymatically desialylating healthy donor SAP made it less active. Some but not all IPF patients, and none of the control patients, had detectable NEU3 in their sera [111]. Mass spectrometry showed that NEU3 desialylates SAP, suggesting that one way that elevated NEU3 in pulmonary fibrosis potentiates fibrosis is by desialylating the endogenous anti-fibrotic protein SAP (Figure 1).

## 6. In Addition to Inactivating SAP, NEU3 Activates 2 Positive Feedback Loops to Potentiate Fibrosis

In human peripheral blood mononuclear cells, NEU3 upregulates extracellular accumulation of the profibrotic cytokines IL-6 and IL-1β, and IL-6 upregulates NEU3, suggesting that NEU3 may be part of a positive feedback loop potentiating fibrosis (bottom left of Figure 1 and [83]). Mammalian sialidases cause extracellular accumulation of active TGF-β1 [82]. Recombinant human LAP is sialylated, and we found that recombinant human NEU3 desialylates LAP [84] causing LAP to release active TGF-β1 ([84] and Figure 2). Conversely, TGF-β1 upregulates NEU3 [112]. Thus, in addition to the NEU3 → IL-6 → NEU3 positive feedback loop, a second positive feedback loop involves NEU3 → TGF-β1 → NEU3 (top left of Figure 1). NEU3 also primes human neutrophils [113], suggesting that upregulated NEU3 can contribute to inflammation in addition to fibrosis.

However, NEU3 may also inhibit TGF-β signaling [114,115]. Human cardiac fibroblasts cultured with TGF-β1 for 72 h have reduced NEU3 mRNA levels and reduced sialidase activity (as assessed with 4-MU-NeuAc at pH 3.8), and overexpression of NEU3 reduced TGF-β1-induced collagen production and inhibited TGF-β signaling [114].

## 7. TGF-β1 Upregulates Translation of *NEU3* and Other mRNAs

We observed increased levels of NEU3 protein in fibrotic lesions in human and mouse pulmonary fibrosis [82]. However, RNA-seq and single-cell RNA-seq indicated that there is no significant increase in the levels of *NEU3* mRNA in IPF patient lungs or lung cell types [116,117]. This suggested that the increased levels of NEU3 protein in fibrotic lesions could be due to increased translation of *NEU3* mRNA. Using sucrose gradients to fractionate RNA-ribosome complexes into monosomes (mRNA with a single ribosome; indicative of a low translation rate per mRNA), and mRNA with multiple ribosomes (polysomes, indicative of high levels of translation) [118], we found that TGF-β1 increases the NEU3 protein in human lung epithelial cells without increasing total levels of *NEU3* mRNA, and causes NEU3 mRNA to shift from monosomes to polysomes [112]. This then indicated that TGF-β1 increases levels of the NEU3 protein by increasing *NEU3* mRNA translation.

## 8. Development of Potent NEU3 Inhibitors That Block Fibrosis in a Mouse Model

In the bleomycin model, the general sialidase inhibitors DANA and Oseltamivir given starting at day 10 strongly decreased fibrosis, with collagen and TGF-β1 levels not significantly different from saline controls [82]. These sialidase inhibitors however have ~10 µM IC50’s for NEU3, and are thus low-potency inhibitors. We found that compounds which mimic the transition state of the sialic acid when NEU3 is removing the sialic acid from a glycoconjugate act as potent (some less than 2 nM IC50) NEU3 inhibitors [84]. We tested three of these NEU3 inhibitors in the mouse bleomycin model of pulmonary fibrosis, and all three (with one showing strong efficacy at 0.1 mg/kg) decreased fibrosis and TGF-β1 accumulation in the lung (supporting the model in Figure 1) when administered starting at day 10, with no discernable toxicity [84]. The observation that the upregulated NEU3 in IPF desialylates and inactivates SAP suggests that our new class of NEU3 inhibitors could be useful as stand-alone therapeutics, or therapeutics to bolster the efficacy of SAP. Sialidase inhibitors may also be useful in other diseases. For instance, in mice, injections of the sialidase inhibitor DANA, or the lack of NEU3 (*Neu3^−/−^* mice) attenuate diet-induced adipose tissue and liver inflammation, and DANA reduces liver steatosis [104,119].

## 9. Development of a Surrogate Biomarker for Blocking NEU3 Activity or Upregulation

We also found that several other glycoproteins are desialylated in the sera of IPF patients [111]. In the mouse bleomycin model, there was also a bleomycin-induced increase in serum protein desialylation at 21 days after bleomycin aspiration. NEU3 inhibitors administered starting at day 10 partially reversed the bleomycin-induced increase in serum protein desialylation. This suggests that in clinical trials of a NEU3 inhibitor, increased serum SAP sialylation, or increased general protein sialylation, could be used as an early surrogate marker for NEU3 inhibitor efficacy.

## Figures and Tables

**Figure 1 ijms-24-00239-f001:**
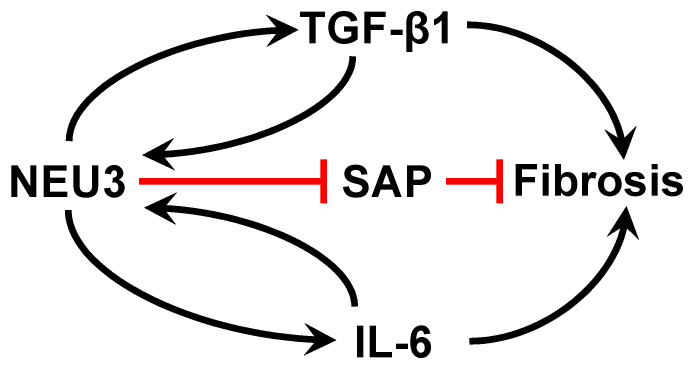
Summary of pathways used by NEU3 to potentiate fibrosis.

**Figure 2 ijms-24-00239-f002:**
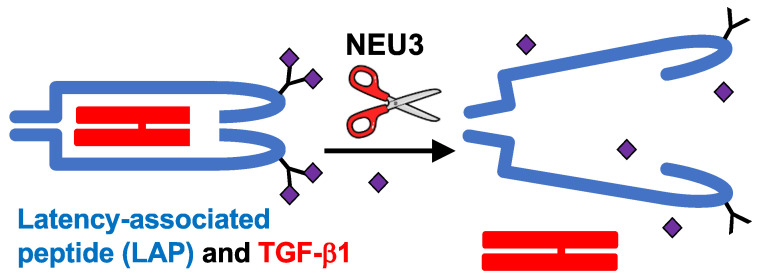
Summary of the NEU3 → TGF-β1 pathway in Figure 1. NEU3 removes sialic acids (purple diamonds) from the LAP protein (blue) that sequesters TGF-β1 (red) in the extracellular environment, causing the LAP to release active TGF-β1.

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
