# Peer review of "Inhibitors of the Sialidase NEU3 as Potential Therapeutics for Fibrosis"

_ijms, 2022, doi:10.3390/ijms24010239_

Round 1
Reviewer 1 Report
The review "Inhibitors of the sialidase NEU3 as potential therapeutics for fibrosis" offers a potential insight into an interesting research topic. The authors should be commended for a comprehensive literature review but I would expect to see far more references/citations from 2020 onwards. (A quick review of the references shows 3 from 2022, 2 from 2021 and 4 from 2020). If more recent references are not available this would raise question marks about the relevance of the topic for a review article.
The structure of the review is not in keeping with the traditional format and I would like to see the topics grouped into common subsections rather than simply numbered 1-9. A comprehensive review requires a better introduction and conclusion to tie in all the topics discussion.
Furthermore, it should be clarified in the body of the review the relevance of the image in the supporting information as it is not clear from the review itself.
Reviewer 2 Report
The review article by Karhadkar et al. summarizes the potential therapeutic role of the sialidase NEU3 inhibitors to treat fibrosis. The authors mainly focus on lung fibrosis, but general observations are made for other organs as well. They offer three valuable mechanisms related to the development of fibrosis by sialidase NEU3 affecting the TGF beta1, SAP and other cytokine (IL-1 and 6) pathways. The markers for sialidase NEU3 from human patients as well as mouse studies provide compelling evidence for the development of inhibitors for sialidase NEU3 as therapeutic targets. These studies come mostly from their own work.
The review is timely, interesting and well written. However, there are some suggestions to change the advertisement tone appearing at some places (see below). Hence, I suggest the authors focus more on the objective science. Line 126-and on: “We formed a company (Promedior) to develop SAP” and Line 131: “Roche bought Promedior and is currently doing the Phase 3 trial”
Line 149-and on: “6. In addition to inactivating SAP, NEU3 activates 2 positive feedback loops to potentiate fibrosis” This is a very important mechanism to describe but it is a one-sided view related to the literature. Can you please comment on and consolidate your findings related to those by Ghiroldi et al. (Biochem J (2020) 477 (17): 3401–3415.) Their findings indicate that “induced up-regulation of sialidase Neu3, a glycohydrolytic enzyme involved in ganglioside cell homeostasis, can significantly reduce cardiac fibrosis in primary cultures of human cardiac fibroblasts by inhibiting the TGF-β signaling pathway, ultimately decreasing collagen I deposition.”
Round 2
Reviewer 1 Report
I am happy to see you have addressed my comments and if the editor agrees to the structure and format of the review I see no reason why it may not be accepted for publication
Author Response
Thank you!